# Long-Term Outcomes of Carbon-Ion Radiotherapy for Malignant Gynecological Melanoma

**DOI:** 10.3390/cancers11040482

**Published:** 2019-04-04

**Authors:** Hiroto Murata, Noriyuki Okonogi, Masaru Wakatsuki, Shingo Kato, Hiroki Kiyohara, Kumiko Karasawa, Tatsuya Ohno, Takashi Nakano, Tadashi Kamada, Makio Shozu

**Affiliations:** 1QST Hospital, National Institutes for Quantum and Radiological Science and Technology, Chiba 263-8555, Japan; murata.hiroto@qst.go.jp (H.M.); okonogi.noriyuki@qst.go.jp (N.O.); kamada.tadashi@qst.go.jp (T.K.); 2Department of Radiation Oncology, Gunma University Graduate School of Medicine, Maebashi 371-8511, Japan; tohno@gunma-u.ac.jp (T.O.); tnakano@gunma-u.ac.jp (T.N.); 3Department of Radiology, Jichi Medical University, Tochigi 329-0498, Japan; waka@jichi.ac.jp; 4Department of Radiation Oncology, Saitama Medical University International Medical Center, Saitama 350-1298, Japan; 5Department of Radiation Oncology, Japanese Red Cross Maebashi Hospital, Maebashi 371-0811, Japan; hiroki.kiyohara@maebashi.jrc.or.jp; 6Department of Radiation Oncology, Tokyo Women’s Medical University School of Medicine, Tokyo 162-8666, Japan; kkarasaw@twmu.ac.jp; 7Department of Reproductive Medicine, Chiba University Graduate School of Medicine, Chiba 260-8677, Japan; shozu@faculty.chiba-u.jp

**Keywords:** carbon-ion radiotherapy, gynecology, melanoma, long-term outcomes, particle beam therapy

## Abstract

Surgical resection is considered a standard therapy for malignant melanoma (MM). However, it has not yet been established as an optimal treatment strategy for gynecological MMs, particularly owing to their very low incidence rates. We retrospectively analyzed clinical outcomes of carbon-ion radiotherapy (C-ion RT) for gynecological MMs. The eligibility criterion was the presence of histologically confirmed gynecological MM. Patients with pelvic or inguinal lymph node metastases were included, while those with distant metastases were excluded. The pelvic and inguinal lymph node regions were irradiated with up to 36 gray relative biological effectiveness (Gy (RBE)) followed by a gross tumor volume boost of up to 57.6 Gy (RBE) or 64 Gy (RBE) in 16 fractions over 4 weeks. Thirty-seven patients (median age: 71 years) were examined. In total, 22 patients had vaginal tumors, 12 had vulval tumors, and 3 had cervical uterine tumors. The median follow-up periods were 23 months (range: 5–103 months) for all patients and 53 months (range: 16–103 months) for survivors. Thirty of 37 patients (81%) achieved complete tumor disappearance. The 2-year local control, overall survival, and progression-free survival rates were 71%, 53%, and 29%, respectively. C-ion RT may be a definitive treatment option for patients with gynecological MM.

## 1. Introduction

Melanomas are malignant tumors arising from melanocytes. Although malignant melanoma (MM) is mostly of cutaneous origin, it can also occur in various extracutaneous sites where melanocytes are present. Mucosal melanomas represent approximately 1.4% of all melanomas [1]. The distribution of head and neck, anal/rectal, female genital tract, and urinary tract mucosal melanomas is 55.4%, 23.8%, 18.0%, and 2.8%, respectively [2]. The vulva is the most frequent site of gynecological MMs (70%), followed by the vagina and, more rarely, the cervix [2,3,4].

The prognosis for patients with gynecological MM is poorer than for those with cutaneous and other types of mucosal non-gynecologic MMs [5,6]. The 5-year overall survival (OS) rate in patients with head and neck, anal/rectal mucosal melanomas, and gynecological melanoma is 31.7%, 19.8%, and 11.4%, respectively [2]. Because of the low incidence of gynecological MM, there is no established optimal treatment to date, although surgery is a common intervention [7]. En bloc excision with a safety margin is thought to be necessary for primary treatment; however, the median age of patients with gynecological MMs is higher than that of patients with other gynecological malignancies [2,3,4]. Thus, not all patients are candidates for surgery because of their advanced age, co-morbidities, or physical condition. Furthermore, even when the tumor is totally resected, the outcomes, in terms of local tumor control and long-term survival, are not satisfactory, show 5-year OS rates of 0–35% [2,3,5,8,9,10,11,12], and may result in postoperative physical and functional disabilities. 

Photon beam radiation therapy (RT) and chemotherapy have a mostly palliative role in patients with MM, which, conventionally, is considered to be a radioresistant tumor that has poor regression after RT. Even the use of high doses per fraction produces a complete remission rate of only 20–30% [13]. Systemic therapy, such as dacarbazine monotherapy or DAV-Feron (dacarbazine, nimustine, vincristine, and interferon-β) therapy, has been used in advanced or recurrent melanoma [14]; however, dacarbazine has never been shown to improve survival in randomized controlled studies [15]. While immune checkpoint inhibitors and molecular targeted drugs were recently tested against cutaneous MMs [16,17], no specific clinical trials to date have validated the effectiveness of these agents against gynecological MMs.

In 1994, the use of carbon-ion (C-ion) RT was initiated at our institute (the National Institute of Radiological Sciences (NIRS)) in Japan. Ion beams, such as protons and C-ions, provide a dose distribution that is superior to that of photons during cancer treatment. Additionally, C-ion beams are heavier than protons and provide a higher relative biological effectiveness (RBE), and, thus, have a higher probability of tumor control, while delivering a smaller dose to the surrounding normal tissue [18,19]. It is therefore reasonable to assume that C-ion RT may be superior to photons for managing tumors characterized by poor radiosensitivity, such as MMs. 

C-ion RT has been shown to produce good local control of MMs in the head and neck regions [20], as well as of choroidal melanoma [21]. In terms of gynecological MM, our group previously reported the preliminary results of a clinical trial comprising 23 patients with this disease who underwent definitive C-ion RT [22]. Given this research history, we conducted a long-term follow-up study of C-ion RT in a larger number of patients with gynecological MM.

## 2. Results

### 2.1. Patient Characteristics

Between January 2004 and December 2017, 38 patients with gynecological MM were treated by C-ion RT using our protocol. One patient had a lung metastasis at the start of C-ion RT and was excluded from the analysis. Thus, 37 patients with gynecological MM were analyzed in this study, and patient and tumor characteristics are listed in Table 1. The patients’ ages were 51–88, with a median age of 71. Twenty-two patients had vaginal tumors, 12 had vulval tumors, and 3 had cervical uterine tumors. Nine of the patients’ tumors were post-surgical recurrences, and three had received chemotherapy prior to C-ion RT. Two patients who had tumors of >60 mL were irradiated with a total dose of 64.0 gray relative biological effectiveness (Gy (RBE)) in 16 fractions, 35 patients were irradiated with a total dose of 57.6 Gy (RBE) in 16 fractions. The median follow-up period was 23 months (range: 5–103 months) for all patients, and 53 months (range: 16–103 months) for those who survived. 

### 2.2. Treatment Efficacy and Prognostic Factors

In terms of initial tumor response (i.e., the maximum reaction within six months after commencing C-ion RT), 19 patients achieved complete response (CR), 14 achieved partial response (PR), and 4 had stable disease. Among the 18 patients who did not achieve CR, 11 eventually did achieve this status after longer treatment with C-ion RT. Thus, 30 of 37 patients (81%) eventually achieved tumor disappearance following C-ion RT. Twenty-five patients had died before the final follow-up date, of which 21 died from MM and 4 died from non-cancer-related reasons (pneumonia, pylethrombosis, myocardial infarction, and subarachnoid hemorrhage). The 2-year local control (LC), OS, and progression-free survival (PFS) rates were 71% (95% confidence interval (CI): 53.6–87.6%), 53% (95% CI: 36.3–69.2%), and 29% (95% CI: 14.0–4.7%), respectively, while the 5-year LC, OS, and PFS were 44% (95% CI: 20.6–68.3%), 28% (95% CI: 12.0–44.8%), and 17% (95% CI: 4.3–29.6%), respectively (Figure 1). The 2-year and 5-year distant metastatic rates were 45% (95% CI: 38.3–71.5%) and 57% (95% CI: 25.5–60.7%), respectively (Figure 2).

Table 2 shows the results of the log-rank tests for the prognostic factors. None of the factors examined (including age, prior treatment, T stage, tumor size, lymph node metastasis, adjuvant therapy, and initial tumor response) significantly influenced LC, PFS, and OS in univariate analysis. However, age was associated with the rate of distant metastasis, where the younger group (age < 71 years) showed a higher incidence of distant metastasis than the elderly group (age ≥ 71 years) (*p* = 0.041). 

### 2.3. Toxicity

The acute and late toxicities observed in all patients are listed in Table 3. In terms of acute toxicity, three patients developed grade 3 dermatitis or mucositis. These toxicities were manageable medically and no other acute toxicities of grade 3 or worse were observed. None of the patients developed late grade 3 or worse toxicities.

## 3. Discussion

Ours was the first study to evaluate the safety and efficacy of C-ion RT for gynecological MM with long-term follow-up. To date, conventional photon RT has not been able to achieve temporal CR because of the radioresistance inherent in gynecological MMs [13]. Meanwhile, the present study demonstrated a favorable local effect of C-ion RT, wherein 30 of 37 patients (81%) eventually achieved tumor disappearance following the treatment. Representative tumor response is shown in Figure 3. Moreover, we found that the 2- and 5-year LC rates were 71% and 44%, which is notable given that our study included medically inoperable elderly patients with a median age of 71 years old. Additionally, no severe late toxicity related to C-ion RT was observed. To date, the standard treatment for gynecological MM is radical surgery with regional lymphadenectomy, procedures which consider lesion size, thickness, and depth of invasion. However, local recurrence is frequent following surgery (occurring in 40–60% of the patients) [23,24], and such treatment procedures also incur postoperative physical and functional disabilities [2,3,4,7]. Thus, C-ion RT for gynecological MM appears to be worthy of consideration as a local therapy.

The 5-year estimated OS rate among our patients was 28%. Previous studies of gynecological MM, which mainly comprised patients who underwent surgery, showed 5-year OS rates of 0–35% (Table 4) [2,3,5,8,9,10,11,12]. The corresponding OS rate of patients who underwent C-ion RT in our study was comparable to previous results. Another previous study showed that tumor diameter appears to be the most predictive survival factor, where tumors <30 mm in size are associated with longer survival [23]. However, tumor size was not a significant survival factor in the univariate analysis in our study. Based on our observed outcomes as well as the low incidence of severe late toxicities in our patients, C-ion RT shows a genuine survival benefit, and may, therefore, be a definitive treatment choice for patient with gynecological MM.

However, the actual OS rates for gynecological MM are far from satisfactory regardless of treatment modalities. A previous study found that the 5-year survival rate for cutaneous MM was approximately 80% [25]. The main reason for poor OS among patients with gynecological MM may be the inherent aggressiveness of the tumor and distant metastasis. Hou et al. reported that gynecological MM has unique molecular features compared to non-gynecological melanoma [26], and that programmed cell death ligand 1 (PD-L1) (56%) and programmed cell death (PD-1) (75%) were frequently expressed in gynecological MM [26]. These findings may explain the high incidence of distant metastasis associated with gynecological melanomas [27]. As shown in Figure 2, patients in the present study developed distant metastases with high frequency, especially within 12 months after C-ion RT. Therefore, concurrent or adjuvant use of anti-PD-L1 or anti-PD-1 drugs with C-ion RT is warranted in future clinical trials. Hou et al. also reported that the *B-Raf* serine/threonine kinase was the most frequently mutated proto-oncogene in gynecological MM, occurring at a rate of 26% compared to 8.3% in patients with mucosal non-gynecological melanoma, whereas phosphatidylinositol 3-kinase pathway mutations and estrogen receptor/progesterone receptor overexpression were rare [26]. Thus, molecular-targeted therapies could also be promising for patients with gynecological MM.

## 4. Materials and Methods

### 4.1. Eligibility

The present retrospective study was conducted using the framework of the Working Group of Gynecological Tumors. This study was approved by our institutional review board—National Institute of Radiological Sciences Certified Review Board (Study ID: NIRS-18-001)—and was conducted in compliance with the Declaration of Helsinki. The board waived the requirement for informed consent owing to the retrospective nature of this study.

The treatment of gynecological MM with C-ion RT commenced at our institution in 2004. Among patients who received C-ion RT between January 2004 and December 2017, those who met the following eligibility criteria were included in the present study: (i) histologically confirmed MM of the gynecological regions, (ii) localized measurable tumors, (iii) lymph node metastasis confined to the inguinal lymph nodes and pelvic region where their irradiation was possible in the same field, (iv) 20 years of age or older, (v) Eastern Cooperative Oncology Group performance status score of 0–2, (vi) refused surgery or were medically inoperable, (vii) no critical complications or active double malignancy, and (viii) expectation of survival for at least 6 months. The exclusion criteria were: (i) tumors with uncontrollable distant metastases, (ii) active intractable infection in an irradiation area, and (iii) prior radiotherapy in an irradiation area. Patients were staged according to the Union for International Cancer Control TNM Classification of Malignant Tumors, 7th edition.

All patients underwent abdominal and pelvic computed tomography (CT), pelvic magnetic resonance imaging (MRI), and ^18^F-fluorodeoxyglucose positron emission tomography (PET)-CT scans for accurate staging. Tumor size was assessed by pelvic examination and MRI.

### 4.2. C-ion RT

With respect to C-ion RT treatment planning, 2.0–2.5 mm-thick CT images were acquired from each patient for 3-dimensional treatment planning. Patients underwent CT in the supine position using customized cradles and were immobilized with a low temperature thermoplastic sheet. The dose calculation was performed using the HIPLAN or Xio-N2 software programs (National Institute of Radiological Sciences, Chiba, Japan). The calculated radiation dose for the target volume and surrounding normal structures was expressed in Gy (relative biological effectiveness (RBE)), which was defined as the physical doses multiplied by the C-ion RBE, using a semi-empirical and modified microdosimetric kinetic model [19,28].

Patients were administered C-ion RT daily for 4 days per week (Tuesday through Friday). Treatment consisted of pelvic irradiation and local boost. The gross tumor volume (GTV) was delineated using MRIs and a clinical examination conducted immediately prior to each planning session. The clinical target volume (CTV)-1 included all areas of gross and potentially microscopic disease, and encompassed the uterus, vagina and/or vulva, pelvic lymph nodes (internal iliac, external iliac, and obturator), and inguinal lymph nodes. The first planning target volume (PTV-1) included the CTV-1 plus a 5–10 mm safety margin for positioning uncertainty. The PTV-1 was irradiated with 36 Gy (REB) in 10 fractions via 3 portals and was covered by at least 90% of the prescribed dose. The CTV-2 was defined as the GTV and GTV node with a minimum margin of 5 mm, which was added to the CTV-2 to produce the PTV-2. A dose of 21.6 Gy (RBE) of C-ion RT in 6 fractions was delivered to the PTV-2 via 2–3 portals. Thus, the total dose to the MM was 57.6 Gy (RBE) in 16 fractions.

Starting in 2011, patients with tumors of >60 mL were irradiated with up to a total dose of 64.0 Gy (RBE) in 16 fractions, which was based on the treatment of head and neck MM [20]. For these patients, the PTV-1 was irradiated with 36 Gy (REB) in 9 fractions and the PTV-2 was irradiated with 20 Gy (REB) in 5 fractions, after which the PTV-3 (consisting of the GTV plus a 3 mm margin) was irradiated with 8 Gy (REB) in 2 fractions. The gastrointestinal tract was excluded from the PTV-3 to limit its exposure to a maximum of 60.0 Gy (RBE).

### 4.3. Patient Preparation for Daily Treatment

At each treatment session, the patient was first positioned on the treatment couch using immobilization devices. Next, the patient’s position was verified using a computer-aided online positioning system. Digital orthogonal radiography images (positioning images) that were acquired and transferred to the positioning computer were compared to reference images that were digitally reconstructed from CT scans. A positioning difference of >2 mm required readjusting the treatment couch until an acceptable position was attained. Furthermore, if gas was detected in the rectum in digital orthogonal radiography images, patients were administered enemas to clear any stool. To minimize the internal motion of target organs, normal saline (100–150 mL) was infused into the bladder, and vaginal packing was performed tightly at each treatment session. The cotton for vaginal packing was soaked in a contrast medium to enable visualization of the vaginal position via radiography. Patients were encouraged to take laxatives to prevent constipation during the treatment period. 

### 4.4. Evaluation

Initial tumor response was defined as the maximum reaction within 6 months after commencing C-ion RT via physical examination, MRI, CT, and ^18^F-fluorodeoxyglucose PET. Tumor response was defined using MRI according to the Response Evaluation Criteria in Solid Tumors Version 1.1 [29]. Acute reactions of normal tissue were classified according to the Cancer Therapy Evaluation Program, Common Terminology Criteria for Adverse Events, Version 4.0 [30], with a maximum reaction occurring within 3 months after initiation of therapy. Late reactions were classified according to the Radiation Therapy Oncology Group/European Organization for Research and Treatment of Cancer scoring system [31]. All patients were evaluated with CT and MRI every 3 months for the first 2 years and every 6 months thereafter. Recurrences were detected by physical examination, CT, MRI, ^18^F-fluorodeoxyglucose PET, and/or biopsy. The effects of the treatment were evaluated in terms of LC, PFS, and OS.

### 4.5. Statistical Analyses

LC, PFS, OS, and distant metastatic curves were plotted using the Kaplan-Meier method. Log-rank, ANOVA post-hoc tests, and univariate analyses were performed using the Statistical Package for the Social Sciences software for Macintosh, version 24.0 (IBM Inc., Armonk, NY, USA). A *p*-value of < 0.05 was considered statistically significant.

## 5. Conclusions

In conclusion, we found that C-ion RT may serve as a definitive treatment choice for patients with gynecological MM. As the retrospective nature of our study is a notable limitation, further prospective studies are warranted to validate the effectiveness of C-ion RT in these patients.

## Figures and Tables

**Figure 1 cancers-11-00482-f001:**
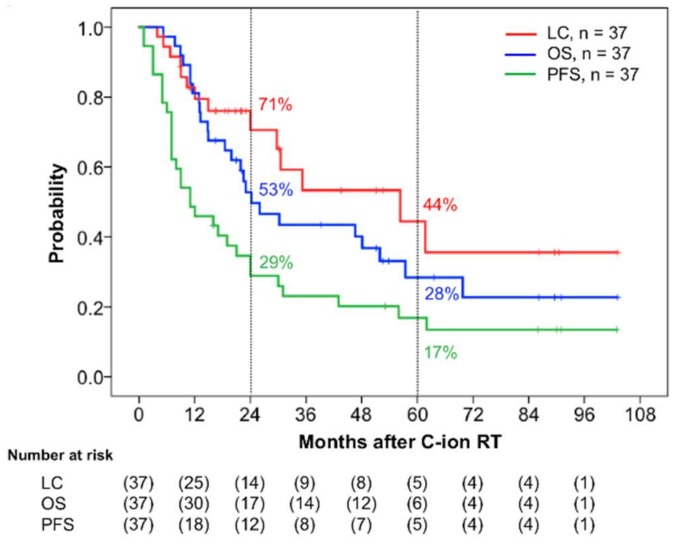
Kaplan–Meier curves of the clinical results. Local control (LC) is shown in red, overall survival (OS) in blue, and progression-free survival (PFS) in green for all 37 patients. The numbers at risk are shown below the figure.

**Figure 2 cancers-11-00482-f002:**
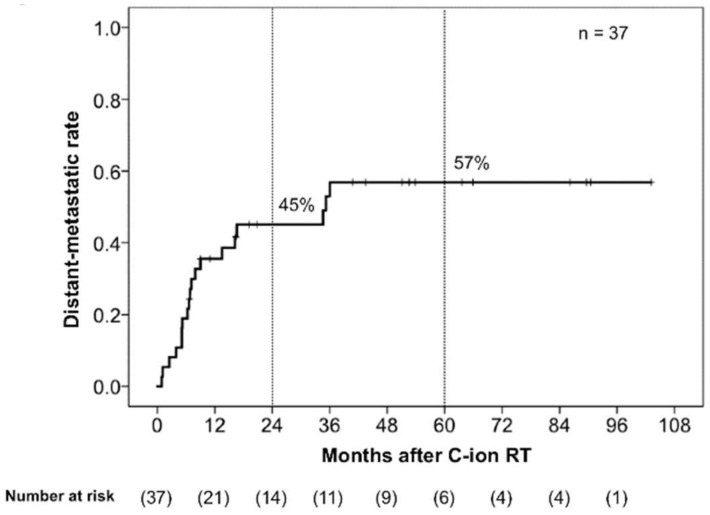
Kaplan–Meier curve of the distant metastatic rates. The numbers at risk are shown below the figure.

**Figure 3 cancers-11-00482-f003:**
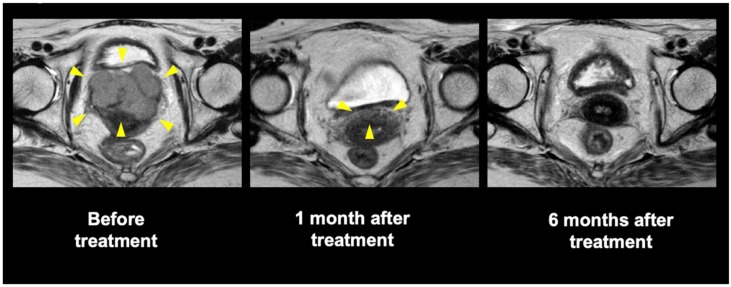
Pelvic magnetic resonance imaging of a representative patient who underwent carbon-ion radiotherapy. Yellow arrow: tumor or remained tumor.

**Table 1 cancers-11-00482-t001:** Patient and tumor characteristics (*n* = 37).

Characteristics	Number of Patients	%
Age (median), years	51–88 (71)	
Tumor site		
Vagina	22	60
Vulva	12	32
Cervix uterus	3	8
Prior treatment		
Surgery	9	24
Chemotherapy	3	8
None	25	68
T stage (including recurrent T stage)		
T1	8	22
T2	21	56
T3	8	22
Tumor size in maximal diameter		
≤30 mm	29	78
>30 mm	8	22
Lymph node metastasis		
Positive	5	14
Negative	32	86
The reason for inoperableness		
Medically inoprerable	27	73
Patient’s refusal	10	27
Total dose of C-ion RT		
57.6 Gy (RBE) in 16 fractions	35	95
64.0 Gy (RBE) in 16 fractions	2	5
Adjuvant therapy		
DAV/DAV Feron	9	24
Nivolumab	1	3
None	27	73

C-ion RT = carbon-ion radiotherapy, RBE = relative biological effectiveness, DAV = dacarbazine, nimustine, and vincristine, Feron = interferon β.

**Table 2 cancers-11-00482-t002:** Assessment of the prognostic factors using univariate analysis.

Factor	No. of Patients	LC		PFS		OS		DM	
		2-Year (%)	*p*-Value	2-Year (%)	*p*-Value	2-Year (%)	*p*-Value	2-Year (%)	*p*-Value
Age (years)			0.213		0.617		0.983		0.041
<71	17	49.7		17.6		57.0		52.9	
≥71	20	89.2		39.4		43.3		40.1	
Prior treatment			0.468		0.547		0.564		0.242
No	12	69.4		30.5		53.5		37.6	
Yes	25	72.2		25.0		50.0		58.3	
T stage (including recurrence)			0.974		0.953		0.877		0.903
T1–2	29	65.4		26.6		53.7		48.0	
T3	8	87.5		37.5		37.5		37.5	
Tumor diameter			0.337		0.418		0.304		0.320
≤30 mm	29	73.9		33.4		57.2		46.3	
>30 mm	8	60.0		12.5		37.5		37.5	
LN metastasis			0.320		0.248		0.069		0.206
Positive	5	60.0		0.0		40.0		80.0	
Negative	32	73.0		40.4		54.9		39.1	
Adjuvant chemotherapy			0.535		0.142		0.382		0.796
No	27	65.8		20.4		53.8		43.4	
Yes	10	80.0		50.0		50.0		50.0	
Tumor response within 6 months after commencing C-ion RT			0.535		0.923		0.818		0.826
CR	19	77.7		23.7		61.5		43.2	
Non-CR	18	61.6		33.3		43.2		45.8	
Primary site			N.S.		N.S.		N.S.		N.S.
Vagina	22	73.4		26.5		55.2		53.0	
Vulva	12	76.4		33.3		58.3		33.3	
Cervix uterus	3	33.3		33.3		33.3		33.3	

No. = number, LC = local control, PFS = progression-free survival, OS = overall survival, DM = distant metastasis, LN = lymph node, C-ion RT = carbon-ion radiotherapy, CR = complete response. N.S. = not significant.

**Table 3 cancers-11-00482-t003:** Acute and late toxicities (*n* = 37).

Acute Toxicity	CTCAE v.4 Scoring				
	Grade 0	Grade 1	Grade 2	Grade 3	Grade 4–5
Dermatitis/mucositis	2	18	14	3	0
Genitourinary toxicity	28	9	0	0	0
Lower gastrointestinal toxicity	17	14	6	0	0
**Late toxicity**	**RTOG/EORTC Scoring**				
	**Grade 0**	**Grade 1**	**Grade 2**	**Grade 3**	**Grade 4–5**
Dermatitis/mucositis	28	9	0	0	0
Genitourinary toxicity	30	3	4	0	0
Lower gastrointestinal toxicity	29	5	3	0	0

CTCAE v.4 = Common Terminology Criteria for Adverse Events, Version 4.0, RTOG/EORTC = Radiation. Therapy Oncology Group/European Organization for Research and Treatment of Cancer.

**Table 4 cancers-11-00482-t004:** Review of previously reported clinical outcomes in patients with gynecological malignant melanoma.

Authors (ref.)	Year	Treatment Modality	Number of Patients	Primary Site	5-Year OS
Bradgate et al. [8]	1990	Surgery	50	Vulva	35%
Look et al. [9]	1993	Surgery	16	Vulva	30%
Ragnarsson-Olding et al. [5]	1993	Mainly surgery	245	Mixed	35% (vulva); 13% (vagina)
Chang et al. [2]	1995	Mainly surgery	59	Mixed	11% as DFS
Clark et al. [10]	1999	Mainly surgery	37	Uterus cervix	14% in Stage II; 0% in Stage III–IV
Verschraegen et al. [11]	2001	Surgery	31	Vulva	31% in Stage II–IV
Irvin et al. [3]	2001	Surgery	16	Vulva	17% months in MST
Frumovitz et al. [12]	2010	Surgery	37	Vagina	20%
Present study	2018	C-ion RT	37	Mixed	28%

ref. = reference, OS = overall survival, DFS = disease-free survival, MST = median survival time, C-ion RT = carbon ion radiation therapy.

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
