# Peer review of "Long-Term Outcomes of Carbon-Ion Radiotherapy for Malignant Gynecological Melanoma"

_cancers, 2019, doi:10.3390/cancers11040482_

Round 1
Reviewer 1 Report
The authors reported clinical outcomes of C-ion RT for gynecological MMs. This treatment modality can achieve good local control and OS including medically inoperable elderly patients. This is a very important and interesting paper I have no hesitation in recommending it for publication following some minor tidy-up.
Here are the minor comments.
How many cases are medically inoperable patients?
It might be better included in table 1.
Only 2 cases were treated with 64.0 Gy (RBE) in 16 fractions.
Why this protocol was canceled?
Page 8 line 236
PET requires use of radioactive contrast agents. What kind of tracers were used?
Author Response
Reviewer 1
Comments and Suggestions for Authors
The authors reported clinical outcomes of C-ion RT for gynecological MMs. This treatment modality can achieve good local control and OS including medically inoperable elderly patients. This is a very important and interesting paper I have no hesitation in recommending it for publication following some minor tidy-up. Here are the minor comments.
Response:
We would like to thank the comments and suggestion, which have helped us improve our manuscript.
How many cases are medically inoperable patients? It might be better included in table 1.
Response:
Among all 37 patients, 27 patients were medically inoperable. We added this information in table 1. Thank you for your suggestion.
Only 2 cases were treated with 64.0 Gy (RBE) in 16 fractions. Why this protocol was canceled?
Response:
We apologize the confusion regarding the description of protocols.
As we described in the original manuscript, starting in 2011, patients with tumors of >60 mL were irradiated with a total dose of 64.0 Gy (RBE) in 16 fractions. Only two patients had tumors of >60 mL, then these two patients were received carbon-ion irradiation at a dose of 64.0 Gy (RBE) in 16 fractions. The strategy is still on-going.
We added these points in the “2.1. Patient Characteristics” as follows;
Page 2, line 88:
Two patients who had tumors of >60 mL were irradiated with a total dose of 64.0 Gy (RBE) in 16 fractions, 35 patients were irradiated with a total dose of 57.6 Gy (RBE) in 16 fractions. The median follow-up periods were…
Page 8 line 236
PET requires use of radioactive contrast agents. What kind of tracers were used?
Response:
We apologize the insufficient description. 18F-fluorodeoxyglucose positron emission tomography (PET) scans were performed. We added this point in manuscript.
Page 8, line 238:
Initial tumor response was determined as the maximum reaction within 6 months after commencing C-ion RT via physical examination, MRI, CT, and 18F-fluorodeoxyglucose PET.
Page 8, line 246:
Recurrences were detected by physical examination, CT, MRI, 18F-fluorodeoxyglucose PET, and/or biopsy.
Reviewer 2 Report
Review for Manuscript Cancers-476157-peer-review-v1
General Comments: Overall, very well written and presentation of the data in figure and table form. Just a few small specific comments by line number below.
More Specific Comments:
Title – None
Abstract – Line 27 – Add “the” after “was”
Introduction
1) Line 45 – Change “pigment cells” to “melanocytes”
2) Line 58-59 – Expand on outcomes and survival rates that make surgical resection un-satisfactory with regard to outcomes and survival rates.
3) Line 66-68 – Nice inclusion of immunotherapy, here, and throughout.
Results – Line 121 – For “were manageable”, should this be “were manageable medically”?
Discussion – Line 158 – Add “and distant metastasis” after “of the tumor”
Methods – None
Conclusions – None
Figures, Tables, and Legends
1) Table 1 – The % column values are being cut off.
2) Figure 3 – Add arrows to help the reader see the tumor margins in all subfigures.
Author Response
Reviewer 2
General Comments: Overall, very well written and presentation of the data in figure and table form. Just a few small specific comments by line number below.
Response:
We would like to thank the comments and suggestion, which have helped us improve our manuscript.
More Specific Comments:
Title – None
Response:
Thank you for your feedback.
Abstract – Line 27 – Add “the” after “was”
Response:
We added the article.
Page 1, line 27:
The eligibility criterion was the presence of histologically confirmed gynecological MM.
Introduction
1) Line 45 – Change “pigment cells” to “melanocytes”
Response:
We changed the term of “pigment cells” to “melanocytes”
Page 1, line 42:
Melanomas are malignant tumors arising from melanocytes. Although malignant melanoma (MM) is mostly of cutaneous origin, it can also occur in various extracutaneous sites where melanocytes are present. Mucosal melanomas…
2) Line 58-59 – Expand on outcomes and survival rates that make surgical resection un-satisfactory with regard to outcomes and survival rates.
Response:
Thank you for your suggestion. We added the OS rates with several references as follows; Relating this changing, we also renumber the references in the manuscript and Table 4.
Page 2, line 56:
Furthermore, even when the tumor is totally resected, the outcomes in terms of local tumor control and long-term survival are not satisfactory, showed 5-year OS rates of 0–35% [2,3,5, 8-12],…
3) Line 66-68 – Nice inclusion of immunotherapy, here, and throughout.
Response:
Thank you for your feedback.
Results – Line 121 – For “were manageable”, should this be “were manageable medically”?
Response:
We changed the description of “were manageable” to “were manageable medically”
Page 4, line 121:
The acute and late toxicities observed in all patients are listed in Table 3. In terms of acute toxicity, 3 patients developed grade 3 dermatitis or mucositis; these toxicities were manageable medically, and no other acute toxicities of grade 3 or worse were observed.
Discussion – Line 158 – Add “and distant metastasis” after “of the tumor”
Response:
We added the phrase.
Page 4, line 160:
The main reason for poor OS among patients with gynecological MM may be the inherent aggressiveness of the tumor and distant metastasis.
Methods – None
Response:
Thank you for your feedback.
Conclusions – None
Response:
Thank you for your feedback.
Figures, Tables, and Legends
1) Table 1 – The % column values are being cut off.
Response:
We changed width of the column.
If it still looks being cut off, please tell us again.
2) Figure 3 – Add arrows to help the reader see the tumor margins in all subfigures.
Response:
Thank you for your suggestion. We revised the Figure 3 (added arrows in the subfigures).

Reviewer 3 Report
My comments are in the attached file.

Author Response
Reviewer 3
The authors address a topic of interest that has not been frequently studied in the literature on radiotherapy, and that provides new and valuable data. The work is well written and cane be followed easily. Therefore, I believe that this work deserves to be published in Cancers.
I have only a few comments to the authors:
Response:
We would like to thank the comments and suggestion, which have helped us improve our manuscript.
Because it is an important issue in carbon-ion radiation therapy, the way in which the doses delivered to patients are expressed should be considered in more detail. In my opinion, the authors should follow the criteria established in some recent references [1,2] to express, together, the physical dose and the dose in terms of equieffective dose (EQD) indicating the approximation and methods used to perform the dose transformation and the evaluation of the EQD.
References:
[1] Vogin G, Wambcrsic A, Pötter R, et al. Concepts and terms for dose/volume parameters in carbon-ion radiotherapy: Conclusions of the ULICE taskforce. Cancer Radiother 2018;22:802-809.
[2] Vogin G, Wambcrsic A, Koto M, et al. A step towards international prospective trials in carbon ion radiotherapy: investigation of factors influencing dose distribution in the facilities in operation based on a case of skull base chordoma.
Radiat Oncol 2019;14:24.
Response:
We appreciate for reviewers insightful and valuable comments. We understand that importance of the standardization in description the way in which the doses delivered to patients are expressed. Regarding the use of the terms of equieffective dose (EQD), it is ongoing issue but consensus has yet to be reached in our institution and Japan. Thus, we used the unit of Gy (RBE) in the present manuscript.
To clarify the way of dose description in the present study, we added a sentence as follows;
Page 7, line 203:
The calculated radiation dose for the target volume and surrounding normal structures was expressed in Gy (relative biological effectiveness [RBE]), defined as physical doses multiplied by the C-ion RBE using a semi-empirical and modified microdosimetric kinetic model [19,28].
Abstract, Page 5 (line 134) and Table 2. It is somewhat confusing how a complete response (CR) rate of 81% is determined, and the way this is indicated in Table 2, with percentages that refer to two groups of patients (CR and Non-CR). I think the authors should clarify this point better in the text. On the other hand, this value of 81% cannot be seen in the Figure 3.
Response:
We apologize the confusion regarding the term of “complete response”. There were our insufficient description. Eighty-one percent indicates the best overall response after longer the treatment. In other words, 81% of patients were achieved complete tumor disappearance once after the treatment. Meanwhile, the term of “CR” in Table 2 indicate the maximum reaction within 6 months after commencing C-ion RT, based on RECIST criteria.
We added precise explanation in the column in Table 2.
Table 2, column:
Tumor response within 6 months after commencing C-ion RT
To avoid the confusion regarding the term of “complete response”, we changed the several relating sentences as follows;
Page 1, line 35:
Thirty of 37 patients (81%) achieved complete tumor disappearance.
Page 3, line 99:
Thus, 30 of 37 patients (81%) eventually achieved tumor disappearance following C-ion RT.
Page 5, line 135:
Meanwhile, the present study demonstrated a favorable local effect of C-ion RT, 30 of 37 patients (81%) eventually achieved tumor disappearance following the treatment.
We also added a sentence in the manuscript for explanation of Figure 3. We apologize the insufficient description.
Page 5, line 137:
Representative tumor response is shown in Figure 3.
Point 3: In section 4.1 the first and last paragraphs contain the same information.
Response:
Thank you for your pointing out. We integrated the redundant information.
Page 7, line 179:
The present retrospective study was conducted through the framework of The Working Group of Gynecological Tumors. This study was approved by our institutional review board: National Institute of Radiological Sciences Certified Review Board (Study ID: NIRS-18-001) and was conducted in compliance with the Declaration of Helsinki. The board waived the requirement for informed consent owing to the retrospective nature of this study.
